# Impact of the social context on the prognosis of Chagas disease patients: Multilevel analysis of a Brazilian cohort

**Ariela Mota Ferreira**[1]*, **Éster Cerdeira Sabino**[2], **Lea Campos de Oliveira**[2], **Cláudia Di Lorenzo Oliveira**[3], **Clareci Silva Cardoso**[3], **Antônio Luiz Pinho Ribeiro**[4], **Renata Fiúza Damasceno**[1], **Maria do Carmo Pereira Nunes**[4], **Desirée Sant' Ana Haikal**[1]

**1** Graduate Program in Health Sciences, State University of Montes Claros, Montes Claros, Minas Gerais, Brazil, **2** Institute of Tropical Medicine, University of São Paulo, São Paulo, Brazil, **3** Federal University of São João del-Rey, Research Group in Epidemiology and New Technologies in Health–Centro Oeste Campus, Brazil, **4** Department of Internal Medicine, Federal University of Minas Gerais, Belo Horizonte, Minas Gerais, Brazil

* arielamota@hotmail.com

## Abstract

The present study aims to investigate how the social context contributes to the prognosis of Chagas disease (CD). This is a multilevel study that considered individual and contextual data. Individual data came from a Brazilian cohort study that followed 1,637 patients who lived in 21 municipalities to which CD is endemic, over two years. Contextual data were collected from official Brazilian government databases. The dependent variable was *the occurrence of cardiovascular events in CD* during the two-year follow-up, defined from the grouping of three possible combined events: death, development of atrial fibrillation, or pacemaker implantation. Analysis was performed using multilevel binary logistic regression. Among the individuals evaluated, 205 (12.5%) manifested cardiovascular events in CD during two years of follow-up. Individuals living in municipalities with a larger rural population had protection for these events (OR = 0.5; 95% CI = 0.4–0.7), while those residing in municipalities with fewer physicians per thousand inhabitants (OR = 1.6; 95% CI = 1.2–2.5) and those living in municipalities with lower Primary Health Care (PHC) coverage (OR = 1.4; 95% CI = 1.1–2.1) had higher chances of experiencing cardiovascular events. Among the individual variables, the probability of experiencing cardiovascular events was higher for individuals aged over 60 years (OR = 1.4; 95% CI = 1.01–2.2), with no stable relationship (OR = 1.4; 95% CI = 0.98–2.1), without previous treatment with Benznidazole (OR = 1.5; 95% CI = 0.98–2.9), with functional class limitation (OR = 2.0; 95% CI = 1.4–2.9), with a QRS complex duration longer than 120 ms (OR = 1.5; 95% CI = 1.1–2.3), and in individuals with high NT-proBNP levels (OR = 6.4; 95% CI = 4.3–9.6). CONCLUSION: The present study showed that the occurrence of cardiovascular events in individuals with CD is determined by individual conditions that express the severity of cardiovascular involvement. However, these individual characteristics are not isolated protagonists of this outcome, and the context in which individuals live, are also determining factors for a worse clinical prognosis.

**Data Availability Statement:** All relevant data are within the manuscript and its Supporting Information files.

**Funding:** The study is supported by the National Institute of Health: P50 AI098461-02 and U19AI098461-06. ECS is the author who receives funding. The funders had no role in study design, data collection and analysis, decision to publish, or preparation of the manuscript.

**Competing interests:** The authors have declared that no competing interests exist.

## Author summary

Chagas disease (CD) is a serious public health problem in Latin America and has a strong social impact worldwide. Up to 30% of the infected people may have cardiac alterations, which are associated with a worse prognosis and with high mortality rates. The occurrence of CD is associated with contexts of social vulnerability. However, no studies have been identified that assessed whether unfavorable social contexts are related to the prognosis and evolution of CD, which is the purpose of our study. We evaluated 1,637 patients with CD who lived in 21 municipalities located in regions to which CD is endemic in Brazil, over a two-year period. Of these people, 12.5% evolved into a worse prognosis. Our study revealed that socio-demographic and clinical characteristics of individuals were not isolated protagonists of the evolution of CD. The context in which individuals lived was also a determining factor of a worse prognosis, including living in municipalities with a smaller rural population, fewer physicians, and a smaller Primary Health Care (PHC) coverage. Thus, we observed that characteristics related to the health care available in the municipalities influenced the evolution of CD. This knowledge has the potential to support health care planning that is more appropriate for the evolution of patients with CD, especially considering poor and remote regions.

## Introduction

Chagas disease (CD) is a serious public health problem in Latin America and one of the main Brazilian medical and social problems. CD represents one of the top four causes of deaths from neglected infectious and parasitic diseases in the world and it is included in the group of infectious diseases classified as neglected [1, 2]. The World Health Organization (WHO) estimates a high concentration of CD patients in Latin America, to which the disease is endemic. In Brazil it is estimated that more than 1,100,000 people are affected by CD [1], which remains a major cause of morbidity, mortality, and disability in several Latin American countries. CD was the leading cause of disability-adjusted lost years of life (DALY) among all neglected tropical diseases, and in this group as well as in general, the Brazilian state of Minas Gerais is cited as having one of the highest age-standardized DALY rates [3].

Most patients with CD remain in the "undetermined chronic form", defined as a persistent asymptomatic infection without cardiac or gastrointestinal tract alterations [4]. However, up to 30% of chronically infected people may develop cardiac alterations, which is the most serious complication of CD [5]. Chagasic cardiomyopathy is associated with a worse prognosis, with higher mortality rates compared to other causes of heart failure [4, 6–8].

The prognosis of CD is still strongly impacted by it being neglected, with important problems related to late diagnosis and lack of opportunity for treatment, such as deaths that result from the lack of timely intervention, especially for the cardiac form of the disease [9]. Previous studies estimated that more than 80% of people with CD worldwide will not have access to diagnosis and continued treatment, which support the high morbidity and mortality rates and the social cost of the disease [4, 10].

Despite the knowledge about contextual influence and social determination in the occurrence of some diseases, little has been investigated about the influence of contexts on their evolution and prognosis, which demands more studies to be developed to understand this issue [10]. The occurrence of CD it is admittedly related to contexts of social vulnerability that have been neglected to varying degrees and perspectives [11]. Addressing this problem requires

urgent responses, with emphasis on specific actions by the healthcare network [12] adjusted to the characteristics of each reality [13].

In Brazil the great territorial extent and diversity, with specificities in the ecological, demographic, social, and economic dynamics of the regions, imply multiple and complex clinical, epidemiological, and operational scenarios [14]. These need to be considered in studies related to CD, although no previous studies on the social context related to the prognosis of CD have been identified.

The present study has the objective to investigate the contribution of the social context to the occurrence of cardiovascular events in CD using multilevel modeling, considering a two-year follow-up cohort with more than 1,600 patients with CD who lived in regions of Brazil to which the disease is endemic.

## Methods

### Ethical approval

Ethical approval was obtained from the relevant ethics committee: National Commission of Ethics in Research (CONEP: 179,685/2012). All subjects agreed to participate and signed the informed consent form prior to the beginning of the study.

### Study design

This is a multilevel study on CD that considered individual and contextual data. The individual data came from a prospective two-year follow-up cohort study entitled SaMi-Trop (Research on Biomarkers of Neglected Tropical Diseases in São Paulo/Minas Gerais). This study, conducted in Brazil from 2013 to 2019, covers 21 municipalities, and has been conducted in a multicenter manner, with the participation of four Brazilian public universities. The contextual data used were extracted from an official database of the Brazilian government, and collected at the municipal level.

### Individual data

The SaMi-Trop methodology has been presented in detail in previous publications [14, 15]. The main points of the methodology of this cohort are described in the following paragraphs.

The study was carried out in 21 municipalities selected for showing a high prevalence of CD. These municipalities belong to two regions to which CD is endemic in the state of Minas Gerais, Brazil: the northern region of the state and the Jequitinhonha Valley region.

Patients older than 18 years were recruited to participate in the study based on their CD self-report, during the execution of electrocardiogram (ECG) exams in 2012 by a Telehealth program, which provides distance support to municipal public health services by providing ECG reports and clinical discussions with university specialists [16].

To date, patients followed in this cohort have undergone two assessments, baseline and follow-up. Baseline consisted of 2,161 individuals. At follow-up, performed two years after baseline, 1,709 individuals were evaluated but 145 of the baseline participants had died, totaling 1,854 individuals eligible to be included in the sample (death is one of the events of interest in the present study). However, 217 individuals were excluded from analysis (161 because they did not have positive serology for the anti-*T. cruzi* antibody and 56 because they did not respond to the dependent variable adopted). Consequently, 1,637 individuals were included in the sample (Fig 1).

Baseline data collection occurred between 2013 and 2014, with interviews with the patients, peripheral blood collection, and ECG exams. Follow-up data collection occurred between

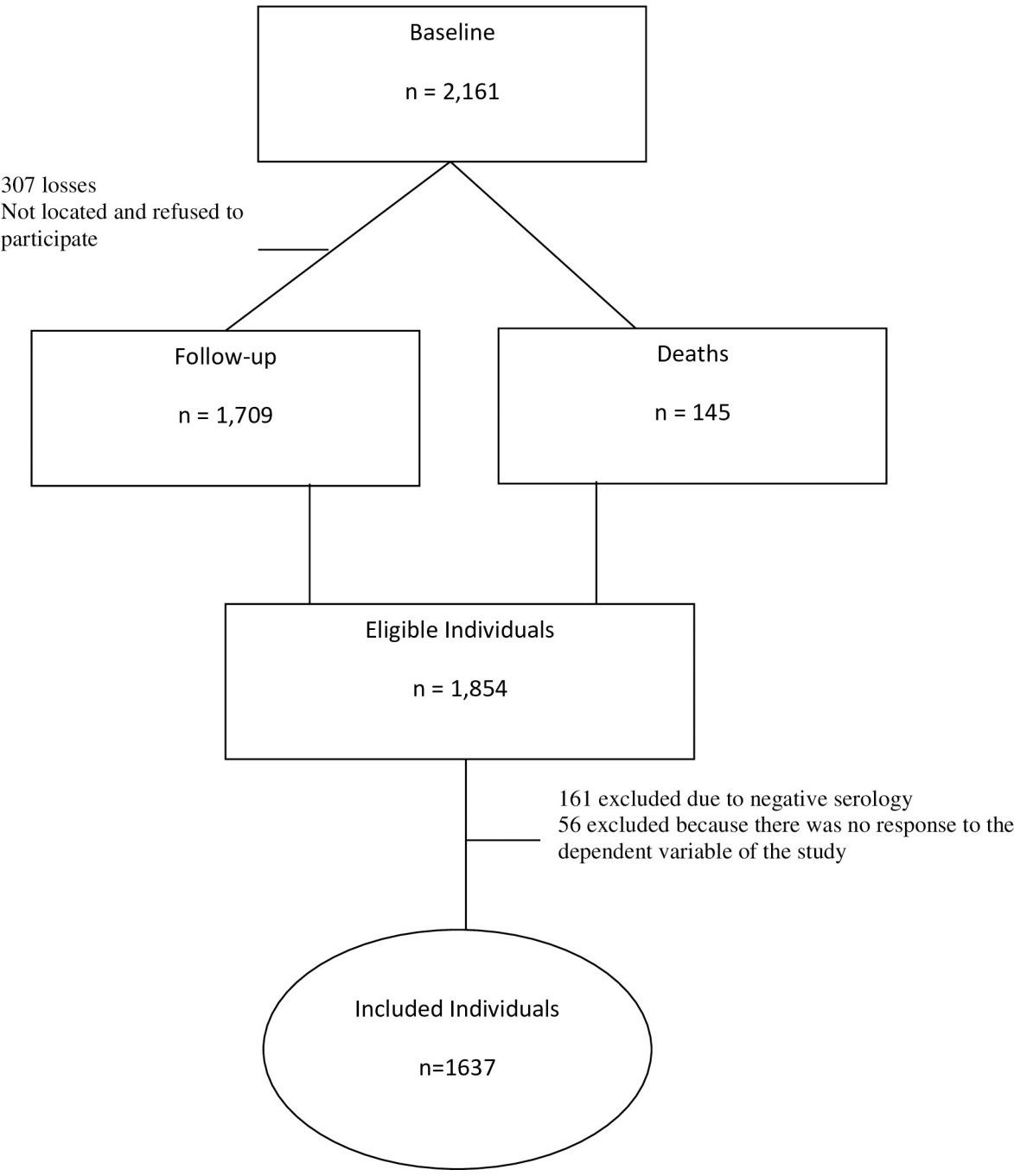

**Fig 1. Flowchart showing the number of eligible, lost, and excluded CD patients in the study.**

2015 and 2016, for which all baseline activities were repeated and the echocardiogram exam was added. The baseline interview included socio-demographic, lifestyle, physical activity, quality of life, and clinical information, in addition to the therapeutic history of CD. In the follow-up interview, data regarding the use of health services, health literacy, and hospitalization were added.

### Contextual data

Contextual data collection was carried out considering the 21 municipalities included in SaMi-Trop. For the social, economic, demographic, epidemiological, and health services characterization of these 21 municipalities, 13 contextual variables were collected from publicly accessible institutional platforms and information systems of the Brazilian government. Table 1 shows these variables, the year adopted as reference for collection (most recent data available), their source, their concept, and the way the data were categorized to carry out the analyses.

### Theoretical model/variables

The organization of variables in the present study followed Andersen & Davidson's [17] conceptual theoretical model, which considers "evaluated health" as an outcome of interest. Following this model, the occurrence of cardiovascular events was adopted as outcome (dependent variable), dichotomized into two categories (absent or present) (Fig 2), considering the two years of follow-up. This variable was constructed from the grouping of three possible events that may have occurred between baseline and follow-up: death (all-cause mortality), obtained from loss of follow-up for this reason and identified by death certificates from the Health Department of Minas Gerais; development of atrial fibrillation (AF) (absent at baseline and present at follow-up) obtained by ECG analysis and defined as sinus rhythm at baseline and AF (presence of irregular trace) on ECG at the follow-up visit; and pacemaker implantation (absent at baseline and present at follow-up) obtained by participants' self-report and confirmed by follow-up ECG through the presence of the image of the pacemaker spike and by ventricular depolarization. The development of the dependent variable including markers that express disease progression aimed to capture changes in the health status of individuals with CD, characterized by over a two-year period. The development of atrial fibrillation and the appearance of atrioventricular blocks, especially total atrioventricular block that requires pacemaker implantation, express progression of heart disease and increased risk of death [18]. To be classified in the category "absent", participants could not have shown new cardiovascular events during the follow-up period.

The independent variables were grouped according to the theoretical model [17] (Fig 2) which has three levels, the first being contextual (first level) and the other two, consisting of individual variables, being individual characteristics (second level) and health-related behavior (third level). Information from the last two levels was extracted from baseline (Fig 2).

In the first level, contextual characteristics related to the municipalities were included considering the variables shown in Table 1, subgrouped into: 1) Predisposing characteristics and 2) Capacitating factors. The variables Municipal Human Development Index (MHDI) and Unified Health System Performance Index (IDSUS) were collected, categorized according to the Brazilian standard, and subsequently dichotomized. The other contextual variables were collected numerically and later dichotomized by adopting the 25th or 75th percentiles as the cutoff point, depending on whether the variable represented a negative measure (low values indicating a better situation) or a positive measure (high values indicating a better situation). The objective was to separate the 25% of the better-off municipalities *vs*. 75% of the worst-off municipalities, given that in general, the municipalities included had similar profiles and most of them had unfavorable social conditions (Table 1).

The second level (individual characteristics) considered three subgroups. The first subgroup was predisposing characteristics: gender (male, female); age (up to 60 years, 60 years or older); self-declared skin color (white, non-white); marital status (stable relationship, no stable relationship); and literacy (yes, no). Age was calculated using the informed date of birth and later dichotomized for the purpose of differentiating adults and the elderly, according to the

**Table 1. Contextual variables collected in publicly accessible institutional platforms and information systems, according to the year, source, concept, and cutoff point adopted in the study.**

| Contextual variables | Collection Year | Source | Concept | Adopted cutoff |
|---|---|---|---|---|
| 1. Total population | 2010 | Atlas of Human Development in Brazil | Population consisting of people living in the municipality | 75th percentile = 31,003 |
| 2. Percentage of the rural population | 2010 | Atlas of Human Development in Brazil | Proportion of the rural population, which covers the whole area outside urban limits | 25th percentile = 33.11% |
| 3. Municipal human development index (MHDI) | 2010 | Atlas of Human Development in Brazil | Geometric average of the dimensions indices: Income, Education, and Longevity, with equal weights | Dichotomized into low *vs*. high and medium, according to the international standard |
| 4. Gini index | 2010 | Atlas of Human Development in Brazil | Measures the degree of inequality in the distribution of individuals according to the per capita household income. Its value ranges from 0 (when there is no inequality) to 1 (when inequality is maximum) | 25th percentile = 0.46 |
| 5. % of the population living in extreme poverty | 2010 | Department of Primary Care–Ministry of Health | Proportion of individuals with a per capita household income equal to or lower than R$ 70.00 per month (U$ 39.54, considering the US dollar exchange rate for January 2010) | 25th percentile = 10.88% |
| 6. Social vulnerability index–SVI | 2010 | Social Vulnerability Atlas | Signals the access, absence, or insufficiency of some civil rights. The three subindices of which it consists are: Urban Infrastructure, Human Capital, and Income/Work | 25th percentile = 0.32 |
| 7. Unified health system performance index (IDSUS) | 2010 | Unified health system performance index | Evaluates the performance of the Unified Health System (SUS) regarding: universality of access, comprehensiveness, equality, resolvability and equity of care, decentralization with single command by management sphere, tripartite responsibility, regionalization, and hierarchization of the health services network | Categorized according to the Brazilian standard and dichotomized into 0.500–0.599 *vs*. 0.600–0.699 and 0.700–0.799 |
| 8. Total health expenditure per inhabitant | 2016 | Public Health Budgets Information System—SIOPS | Measures the total public health expenditure per inhabitant | 75th percentile = R$ 610.72 (U$ 150.79), considering the dollar exchange rate in Jan 2016 |
| 9. Number of doctors per thousand inhabitants | 2017 | National Register of Health Establishment—CNES | Number of doctors present in the municipality per thousand inhabitants | 75th percentile = 0.79 |
| 10. Presence of cardiologists | 2017 | National Register of Health Establishment—CNES | Number of cardiologists present in the municipality hired by the SUS. | 75th percentile = 1 (present *vs*. absent) |
| 11. Number of existing electrocardiographs in SUS facilities per thousand inhabitants | 2017 | National Register of Health Establishment—CNES | Number of electrocardiographs present in the municipality to be used by the SUS per thousand inhabitants | 75th percentile = 0.21 |
| 12. Percentage of the population with health insurance | 2017 | Department of Primary Care—Ministry of Health | Proportion of the population with health insurance | 75th percentile = 3.03% |
| 13. Family Health Strategy (FHS) coverage | 2017 | Department of Primary Care—Ministry of Health | Percentage of the population coverage by Family Health Strategy teams. | 75th percentile = 100% |

*SUS = public health model currently in force in Brazil

Sources: Atlas of Human Development http://www.atlasbrasil.org.br/2013/en/o_atlas/idhm/

Department of Primary Care—Ministry of Health: http://dab.saude.gov.br/portaldab/. Technical Note for October 2017.

Atlas of Social Vulnerability: http://ivs.ipea.gov.br/index.php/en/

Unified Health System Performance Index: http://idsus.saude.gov.br/

SIOPS—Public Health Budget Information System: http://siops-sp.datasus.gov.br/CGI/deftohtm.exe?SIOPS/serhist/municipio/indicMG.def

CNES—National Register of Health Establishment: http://cnes.datasus.gov.br/

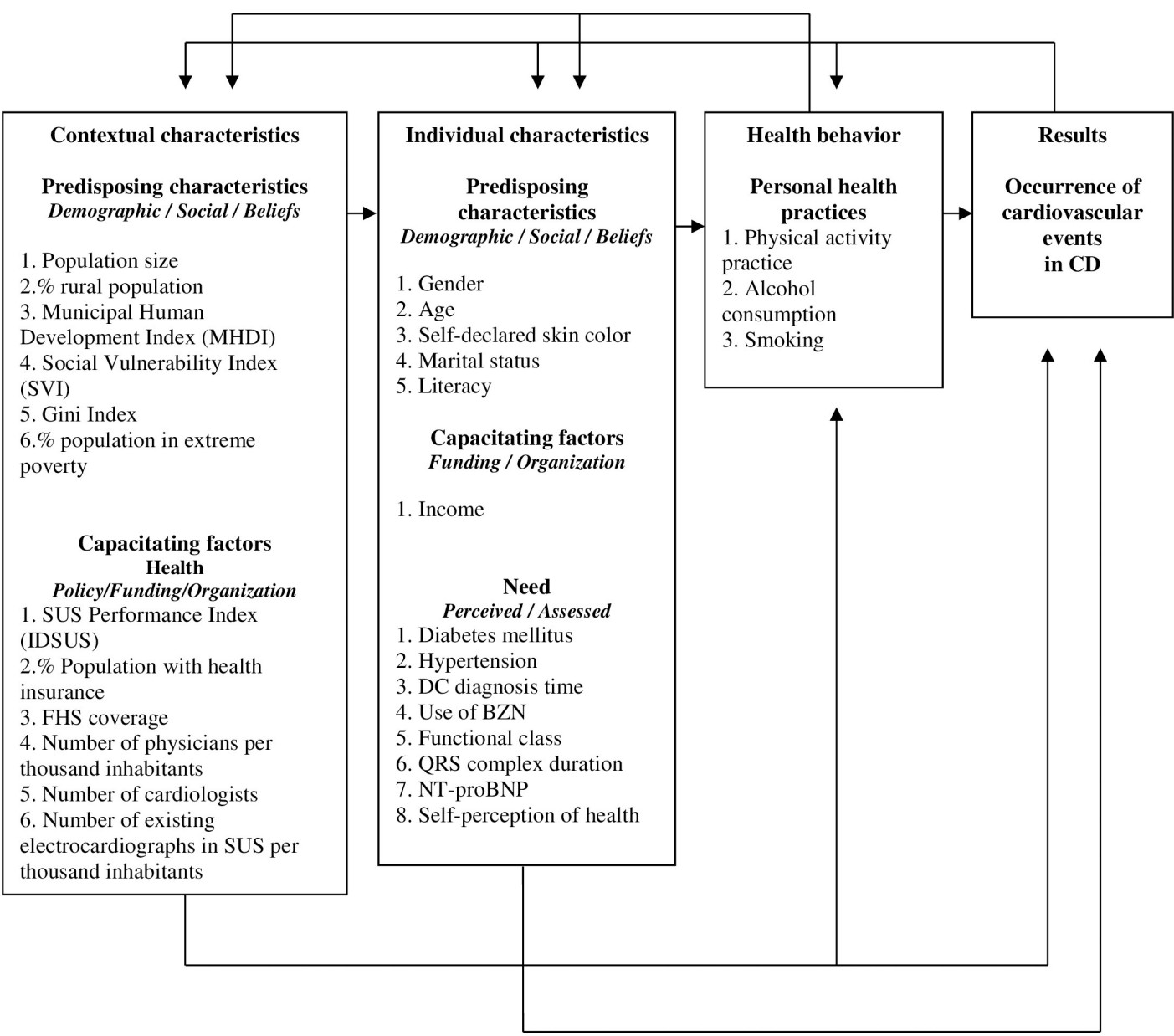

**Fig 2. The adopted behavioral theoretical model.**

criteria adopted by WHO for developing countries [19]. The second was capacitating factors: income (above one minimum wage, up to one minimum wage), dichotomized considering the value of the minimum wage in force in the country (R$ 724.00—U$ 304.20) at the time of data collection. The third subgroup was perceived/evaluated needs: self-reported diabetes diagnosis (no, yes); self-reported hypertension diagnosis (no, yes); self-reported CD diagnostic time (up to ten years, over ten years); use of benznidazole (BZN) sometime in life (yes, no); and functional class (no limitations—class I, with limitations—class II, III, and IV) [20]. The QRS complex duration (up to 119 ms, longer than or equal to 120 ms) [21] and NT-proBNP categorized by age [22] (normal, abnormal) were obtained from ECG examination and blood samples,

**Table 2. Descriptive and bivariate analysis of individual socio-demographic, lifestyle, and health condition-related characteristics and their association with the occurrence of cardiovascular events over two years in Chagas Disease (CD) patients. Minas Gerais, Brazil (n = 1,637).**

| Characteristics | Descriptive analysis | Bivariate analysis | | p-value[π] |
|---|---|---|---|---|
| | | Cardiovascular events | | |
| | | Absent | Present | |
| | n (%) | n (%) | n (%) | |
| *Individual* | | | | |
| Gender | | | | |
| Male | 547 (33.4%) | 459/1432 (32%) | 88/ 205 (42.9%) | 0.002[¥] |
| Age | | | | |
| Up to 60 years | 876 (53.5%) | 805/1432 (56.2%) | 71/205 (34.6%) | <0.001[¥] |
| Self-reported skin color* | | | | |
| White | 349 (21.4%) | 296/1426 (20.7%) | 53/202 (26.2%) | 0.076[¥] |
| Marital Status* | | | | |
| Stable relationship | 1048 (64.3%) | 936/1428 (65.5%) | 112/203 (55.1%) | 0.004[¥] |
| Literate* | | | | |
| Yes | 899 (55.2%) | 816/1427 (57.1%) | 83/203 (40.8%) | <0.001[¥] |
| Income* | | | | |
| Above one minimum wage | 786 (48.2%) | 684/1428 (47.8%) | 102/203 50.2%) | 0.531 |
| Diabetes mellitus | | | | |
| No | 1481 (90.5%) | 1294/1432 (30.3%) | 187/205 (91.2%) | 0.696 |
| Arterial hypertension* | | | | |
| No | 582 (35.6%) | 528/1432 (36,8%) | 54/205 (26.3%) | 0.003[¥] |
| CD diagnosis time* | | | | |
| Up to ten years | 278 (21.9%) | 248/1112 (22.3%) | 30/157 (19.1%) | 0.365 |
| BZN use* | | | | |
| Yes | 410 (27.1%) | 384/1337 (28.7%) | 26/176 (14.7%) | <0.001[¥] |
| Functional class* | | | | |
| No limitations | 904 (55.7%) | 832/1420 (58.6%) | 72/203 (35.4%) | <0.001[¥] |
| QRS complex duration* | | | | |
| Up to 119 ms | 927 (58.2%) | 845/1396 (60.5%) | 82/198 (41.4%) | <0.001[¥] |
| NT-proBNP* | | | | |
| Normal | 1435 (88%) | 1313/1425 (92.1%) | 122/205 (59.5%) | <0.001[¥] |
| Health self-perception* | | | | |
| Positive | 1408 (86.8%) | 1236/1419 (87.1%) | 172/203 (84.7%) | 0.350 |
| *Health Behavior* | | | | |
| Practice of physical activity* | | | | |
| Yes | 374 (23%) | 340/1421 (23.9%) | 34/205 (16.5%) | 0.020[¥] |
| Alcohol consumption* | | | | |
| Does not consume alcohol frequently | 1594 (97.9%) | 1394/1426 (97.7%) | 200/202 (99%) | 0.243 |
| Smoking* | | | | |
| Never smoked or former smoker | 1513 (92.8%) | 1328/1428 (92.9%) | 185/202 (91.5%) | 0.466 |

* Variation of n = 1,637 because of missing information.

[π] Pearson's chi-squared test

[¥] $p \leq 0.20$

respectively. The assessment of self-rated health was based on the question: "How would you rate your health today?" and a Likert scale was adopted with the response options and then dichotomized as positive (good, very good, and medium) *vs.* negative (bad and very bad).

The third level (health behavior) considered only one subgroup related to personal health practices: physical activity practice (yes, no); alcohol consumption (infrequent use of alcohol, frequent use of alcohol); and smoking (never smoked or former smoker, smoker). Data about the practice of physical activity were not changed after collection. Alcohol consumption was measured by asking the question "how many times have you consumed alcoholic beverages in the past thirty days?" with the answer options being: have not consumed, consumed less than once per week, consumed one to two times per week, consumed three to five times per week, and consumed every day. The answers to this question were dichotomized and grouped into two categories: infrequent use (have not consumed/consumed less than once a week/consumed one to two times a week) *vs*. frequent use (consumed three to five times a week/consumed every day). Smoking was evaluated by asking the question: "Which of the following phrases best defines your smoking habits?" with the answer options being: "I have never smoked", "I have smoked but no longer smoke", and "I am currently a smoker". Smokers were considered as those who had the habit of smoking at the time of data collection, and former smokers and those who had never smoked were included in the non-smokers category.

## Statistical analyses

Analysis was carried out to assess differential loss that is, comparing the characteristics of study participants with the characteristics of individuals who were lost and/or excluded. The objective of this step was to verify whether the individuals who continued to be analyzed showed comparability to those lost/excluded. The compared characteristics were gender, self-declared skin color, literacy, age, and income. To perform this type of analysis, descriptive data were obtained and bivariate analyses (chi-square tests) were conducted.

Descriptive analysis of all individual variables involved was subsequently performed. Absolute (n) and relative (%) frequencies were estimated. The outcome was explored, and its frequency was estimated for each municipality included in the present study. Bivariate analysis was subsequently carried out using Pearson's chi-square test. Variables with p value ≤ 0.20 were selected for the multivariate model. Before beginning the multiple analysis, absence of multicollinearity between the independent variables (correlation lower than 0.7) was confirmed. In multivariate analysis, multilevel binary logistic regression was adopted so the variables were introduced in the model by grouping levels, according to the adopted theoretical model. The effects of individual and contextual characteristics on the outcome were analyzed using multilevel models. The multilevel analysis used the fixed effects model (intercept model). Odds ratios (OR) with 95% CI were calculated to assess associations between the outcome and the individual and contextual variables. The model was adjusted upon the introduction of each level, in a hierarchical manner, keeping only variables with statistical significance. Deviance was the indicator used to assess the adjustment quality, making it possible to compare likelihood functions, and is represented by the "-2 loglikelihood". Analyses were run using Predictive Analytics Software (PASW/SPSS) version 18.0 for Windows and STATA version 14.0 (StatCorp, College Station, Texas, USA) statistical software.

## Results

Among the 1,637 patients with CD evaluated in the SaMi-Trop cohort, 205 (12.5%) showed new cardiovascular events between baseline and follow-up, of whom 134 (8.2%) died; 28 (1.7%) developed AF, and 43 (2.6%) required pacemaker implantation. The variation in the occurrence of these events according to each municipality investigated can be seen in Fig 3.

Differential loss analysis showed that, except for the variable age, the other tested variables did not differ significantly in the group of lost individuals when compared with data of those

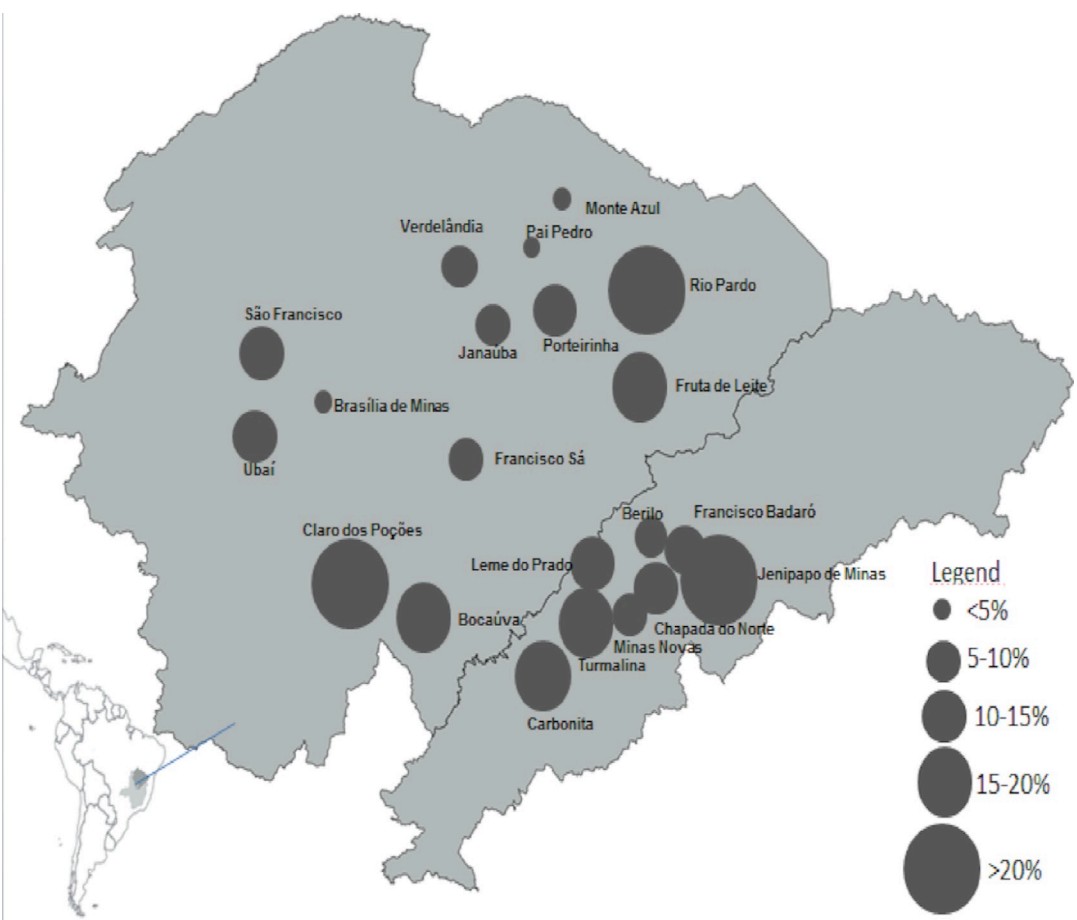

**Fig 3. Occurrence of cardiovascular events over two years in Chagas Disease (CD) patients and their distribution by municipality (n = 21).** Minas Gerais, Brazil. Created with QGIS.

who remained in the study (p > 0.05). However, among the lost individuals there was a significantly higher proportion of older people (55% *vs*. 45%). The mean age in the group of people who remained in the sample was 58.6 (± 12.6) years, while in the group of lost individuals it was 61 (± 13.6) years (p < 0.05). Additionally, in the group of lost individuals with negative and inconclusive serology (n = 161), it was identified 10 (6%) individuals that manifested cardiac events (8 deaths, 1 developed AF, and 1 implanted a pacemaker).

The distribution of participants according to individual characteristics and health behaviors can be seen in Table 2. There was a predominance of individuals under 60 years (53.5%), women (66.6%), non-white self-reported skin color (78.6%), and people with a monthly income of up to one minimum wage (51.8%). The mean age of the individuals was 59.81 (±12.3) years and 65.33 (±13.0) years, among those who did not have, and had cardiovascular events, respectively.

In bivariate analysis, the individual variables selected to make up the initial multiple model (p ≤ 0.20) were: gender, age, self-reported skin color, marital status, literacy, hypertension, BZN use, functional class, QRS complex duration, NT–proBNP, and physical activity practice (Table 2).

The adjusted multiple model revealed that three contextual variables influenced the outcome. Among the contextual characteristics, there was protection for cardiovascular events

**Table 3. Final hierarchical multilevel logistic regression model for the factors associated with the occurrence of cardiovascular events over two years in patients with Chagas disease. Minas Gerais, Brazil (n = 1,637).**

| MODELS | VARIABLES | | OR (CI95%) | p value |
|---|---|---|---|---|
| **Empty model** Deviance (-2log Log likelihood) = 123.406 | | | | |
| **Level 1** *Contextual characteristics* | Rural population | | | |
| | | Smaller rural population | 1 | |
| | | Larger rural population | 0.509 (0.359–0.721) | **<0.001** |
| | | | | |
| | Number of physicians per thousand inhabitants | | | |
| | | Higher number of physicians | 1 | |
| | | Lower number of physicians | 1.698 (1.157–2.490) | **0.007** |
| | | | | |
| | FHS coverage | | | |
| | | Higher FHS coverage | 1 | |
| | | Lower FHS coverage | 1.468 (1.037–2.079) | **0.030** |
| Deviance (-2log Log likelihood) = 121.810 | | | | |
| **Level 2** *Contextual characteristics* **Individual characteristics** | Age | | | |
| | | Up to 60 years | 1 | |
| | | 60 years or over | 1.474 (1.010–2.151) | **0.044** |
| | | | | |
| | Marital status | | | |
| | | Stable relationship | 1 | |
| | | Not in a stable relationship | 1.420 (0.987–2.043) | **0.058** |
| | | | | |
| | Use of benznidazole | | | |
| | | Yes | 1 | |
| | | No | 1.599 (0.985–2.956) | **0.057** |
| | | | | |
| | Functional class | | | |
| | | No limitations | 1 | |
| | | With limitations | 2.007 (1.402–2.873) | **<0.001** |
| | QRS complex duration | | | |
| | | <120 ms | 1 | |
| | | >120 ms | 1.583(1.095–2.289) | **0.014** |
| | Age-adjusted NT-proBNP level | | | |
| | | Normal | 1 | |
| | | Abormal | 6.424 (4.297–9.603) | **<0.001** |
| Deviance (-2log Log likelihood) = 87.861 | | | | |

among those who lived in municipalities with the largest rural population; and higher chances of cardiovascular events among those who lived in municipalities with fewer physicians per thousand inhabitants and those who lived in municipalities with lower Family Health Strategy

(FHS) coverage. In the second level of the individual characteristics there was a higher probability of cardiovascular events among the people who were over 60 years old, did not have a stable relationship, had not used BZN, belonged to a worse functional class, had a QRS complex duration higher than 120 ms, and showed an abnormal age-adjusted NT-proBNP level. No third level variables remained in the model after adjustment (Table 3).

## Discussion

The present study showed that more than 12% of the patients with CD had cardiovascular events over the two-year follow-up. This outcome was associated with the contextual variables: rural population, number of physicians per thousand inhabitants, and FHS coverage; and the individual variables: age, functional class, QRS complex duration and NT-proBNP level. The BZN use and marital status variables showed a borderline association with the outcome and were also maintained in the final model. The high incidence of individuals showing progression of the disease over two years (12%) corroborates the literature regarding the severity of the cardiac form of CD [4, 23]. One may also suspect the difficulty of accessing a quality clinical evaluation in these remote regions.

The methodological approach of the present study goes beyond the individual level of understanding of the health-disease process and reaches the population level, making it possible to grasp the essence of the collective and social character of epidemiology [24]. Previous studies that have adopted a multilevel methodology related to the prognosis of patients with CD have not been identified, making comparisons of this nature impossible.

Regarding the comparative analyses of the groups of patients kept and lost in the present study, it is known that the latter during the follow-up period may differ from those who remain. Individuals who are lost often are those showing the highest proportion of worst socio-demographic indicators, which may represent risk factors relevant to the study [25]. To address this issue, an analysis was carried out to test the presence of this type of bias in the sample. It was observed that there were no significant differences for most of the socio-demographic variables considered in the comparison of the groups, suggesting that they were relatively homogeneous.

The dependent variable was innovatively developed by combining three important events that mark the progression of heart disease: death, development of AF, and pacemaker implantation. Death, the most serious event, is one of the most commonly used health status indicators, especially in studies on health and social inequality [26]. In areas to which CD is endemic, the illness is a leading cause of death from cardiovascular disease [27]. A meta-analysis identified that CD is statistically associated with high mortality rates, regardless of the clinical condition, with a relative risk of 1.74 (95% CI 1.49–2.03) and attributable risk of 42.5% considering the exposed group [28]. AF is associated with an unfavorable prognosis [29, 30]. Previous studies, including a meta-analysis, showed that AF has an independent prognostic value for death, with an OR ranging from 1.14 to 2.8 [30,31]. Pacemaker implantation also represents an important event resulting from chronic Chagas heart disease, which is the most important cardiac consequence of CD [4, 30]. The prevalence of pacemakers among patients with CD has been reported by few studies, ranging from 6.2 to 14.3% [14, 32]. Patients with CD are 13 times more likely to have a pacemaker implanted when compared to people in the general population [4]. In the present study, an incidence of 2.6% of pacemaker implantation was identified in patients with CD over the two-year follow-up. Previous studies showing the incidence of pacemaker implantation in patients with CD were not found.

In the present study, individuals living in municipalities with a larger rural population showed protection against cardiovascular events. The hypothesis proposed by the authors is

that this can be explained by the expansion of the Primary Health Care (PHC) coverage through the FHS, especially in smaller and rural municipalities [33]. The FHS represents the "gateway" to the Unified Health System (*Sistema Único de Saúde*—SUS, the public health model currently available in Brazil [34]. With the expansion mentioned, residents and health workers in more rural areas are more likely to know each other, have a stronger bond, and undertake follow up more closely, facilitating access to information and the scheduling of appointments and tests. In addition, the Brazilian policy of permanent education of SUS workers [35] has aimed to offer training to SUS professionals according to their demands, which originate in the realities that they experience in their practice in health services.

Individuals living in municipalities with fewer physicians per thousand inhabitants had a higher probability of experiencing cardiovascular events. This indicator in isolation may have little significance. Therefore, the WHO does not establish the numerical ratio of physicians per thousand inhabitants considered adequate, as this number depends on regional, socioeconomic, cultural, and epidemiological factors, and consequently it is not possible to establish a generalized "ideal rate" for all countries [36]. Despite these limitations, this indicator is the most used because of the absence of others that encompass the complexity of current care models.

Brazil still has one of the lowest rates of physicians per thousand inhabitants, and in January 2018 this number was 2.18 physicians per thousand inhabitants. There are significant inequalities in the distribution of physicians throughout the Brazilian territory, with the state capitals accounting for 23.8% of the population and 55% of the physicians. The ratio calculated for the set of capitals is 5.07 physicians per thousand inhabitants. In municipalities located in the interior of states, this ratio decreases to 1.28 [37]. The municipalities examined in the present study had an average of 0.68 (± 0.4) physicians per thousand inhabitants, which is almost three times lower than the Brazilian average and half the average calculated for municipalities in the interior of the states, which shows that the analyzed area is economically disadvantaged and has an insufficient healthcare structure, even when Brazilian data is used as a reference. The group of municipalities considered as having the highest number of physicians had an average of 1.21 (± 0.4) physicians per thousand inhabitants, and the group with the lowest number of physicians had an average of 0.51 (± 0.1). Consequently, it must be emphasized that this difference in the number of physicians in the municipalities was important in the occurrence of cardiovascular events. This finding stresses the importance of this marker and provides resources to hypothesize that there will be no improvement in the prognosis of CD if there are no public investments in maintaining more medical professionals in these municipalities, whose health sector can be considered, in general, neglected. No previous studies that considered the number of physicians per thousand inhabitants with outcomes measured at the individual level were identified.

Similarly, individuals living in municipalities with a lower FHS coverage had higher chances of experiencing cardiovascular events. The result obtained in the present study agrees with the literature that points PHC, represented by the FHS, as being associated with better health outcomes [38]. Studies show that PHC-oriented countries have better health indicators, such as lower early mortality caused by preventable causes and longer life expectancy. These countries also encourage population empowerment and provide support to reduce vulnerabilities [39, 40, 41]. Thus, this finding shows the importance of PHC in facing Brazilian health realities, which are extremely heterogeneous and with historical social inequalities, even in the face of commonly neglected health conditions.

There is evidence that social inequalities are strong determinants of a population's health [42]. Individuals literally incorporate the world where they live, producing health, illness, disability, and death standards [24]. According to the WHO, most health illnesses and inequities

occur as a consequence of the so-called "social determinants of health," a term that brings together social, economic, political, cultural, and environmental health issues. The social determinants most commonly associated with the occurrence of diseases are those that generate social stratification and are entitled structural determinants [42]. However, among the several contextual variables tested in the present study, the three that showed association with the outcome of cardiovascular events incite issues that relate more to care than to structural determinants. Structural contextual factors are risk factors for the occurrence of diseases [42], but care contextual factors were more significant in assessing the disease prognosis. This finding is fundamental for a better understanding that care plays a leading role compared to structural issues once the disease is present and taking into account its progression. Proper management of these care determinants must complement individual-level interventions for healthcare professionals to achieve greater effectiveness in CD-targeted actions. It should be noted that the present study did not assess any factor related to access or the quality of care provided.

The individual demographic variables associated with cardiovascular events were marital status and age, corroborating previous studies [43, 44, 45, 46]. Although in our study marital status showed a borderline association with the outcome ($p$ = 0.058), the maintenance of this variable improved the fit of the final model. Furthermore, marital status has been found to play a significant role in adult mortality in previous studies, with married individuals tending to have longer life expectancy when compared to divorced/separated people, widow(er)s, or people who never married [43, 44]. Older individuals were also more likely to have cardiovascular events. Age is widely recognized as an independent factor associated with worse cardiac health and death [4, 29, 30, 45, 46].

The individual clinical variables that remained in the final model are well established and known to be related to the prognosis of the disease [22, 47, 48, 49, 50]. It has been found that more advanced functional class is associated with death caused by increased myocardial dysfunction [47]. Not using BZN also increased the probability of experiencing cardiovascular events, but the $p$ value observed in this association was borderline ($p$ = 0.057) and this result needs to be analyzed with caution. Previous use of this drug is still considered reduced among patients with CD in Brazilian endemic regions (27%) [15]. Previous studies have found that BZN use has been associated with a significant reduction in parasitemia [32], lower prevalence of severe cardiomyopathy markers, and lower mortality [48]. The age-adjusted abnormal NT-proBNP level was the factor most strongly associated with the occurrence of cardiovascular events in the present study. NT-proBNP levels are accurate discriminators of the diagnosis of heart failure, powerful predictors of death, and aid in patient risk stratification [50]. Prolonged QRS complex duration was also associated with the outcome, corroborating a study that identified its prolonged duration as an independent predictor of death in CD [49].

The longitudinal evaluation of a large sample of CD patients who lived in endemic areas and small municipalities, far from the large urban centers commonly depicted in studies in the literature, stands out as one of the strengths of the present study. This allows to extend the results to similar locations, given that the populations with CD usually have a similar epidemiological profile [51]. The creation of a dependent variable considering three cardiovascular events simultaneously is innovative and increases the understanding of factors that may influence the prognosis of CD, unlike other studies that adopt only one event of interest. Results were reliably measured, reflecting the patients' clinical condition as well as their parasitological status.

## Study limitations

An important cardiovascular event assessed in this study was death. Although we acknowledge that the use of all cause mortality is a limitation, in general deaths associated with CD are due

to cardiovascular causes, mainly sudden death or progressive heart failure. In the present study, only 4 non-cardiovascular deaths occurred (one accidental death, two due to cancer and one non-specified death). However, as we were unable to assess the cause of death of each patient, all cause mortality was defined as the outcome.

In the present study the patients were dichotomized according to NYHA functional class into good exercise capacity (Class I) versus others, which may include patients who have different exercise tolerance in the same classification. However, functional class is a subjective estimate of a patient's functional ability based on symptoms that do not always correlate with the objective measures of functional capacity.

Some collected information originated in self-reporting, which may result in measurement bias. However, high accuracy of self-reported data for chronic conditions has already been verified [52]. Additionally, investigating the effect of context on an individual outcome related to the occurrence of cardiovascular events in CD is important, useful, and necessary, because it reveals a reality that is often neglected in many spheres, paving the way for targeted actions and future investigations.

The occurrence of cardiovascular events in CD over two years can be considered high (12.5%) and related to the limitations of the organization/provision of the Brazilian public health service and the organization of the urban/rural space of populations with CD, in addition to socio-demographic and clinical issues already established in the literature. The findings showed that individual conditions are not isolated protagonists in the occurrence of cardiovascular events and that the context in which individuals live also determines this prognosis. The absence of public policies that take into account the context in the health condition of patients with CD can contribute significantly to the high morbidity and mortality in CD. Appropriate investments to expand health care for people with CD in remote and neglected areas need to be made.

## Supporting information

**S1 Checklist. STROBE checklist.**
(DOCX)

**S1 Table. Database.**
(XLS)

## Acknowledgments

We would like to thank all of the SaMi-Trop patients and the health teams in each municipality for their valuable contributions to this study.

## Author Contributions

**Conceptualization:** Ariela Mota Ferreira, Éster Cerdeira Sabino, Lea Campos de Oliveira, Antônio Luiz Pinho Ribeiro, Maria do Carmo Pereira Nunes, Desirée Sant' Ana Haikal.

**Data curation:** Ariela Mota Ferreira, Éster Cerdeira Sabino, Cláudia Di Lorenzo Oliveira, Clareci Silva Cardoso, Antônio Luiz Pinho Ribeiro, Renata Fiúza Damasceno, Maria do Carmo Pereira Nunes, Desirée Sant' Ana Haikal.

**Formal analysis:** Ariela Mota Ferreira, Éster Cerdeira Sabino, Clareci Silva Cardoso, Antônio Luiz Pinho Ribeiro, Renata Fiúza Damasceno, Maria do Carmo Pereira Nunes, Desirée Sant' Ana Haikal.

**Funding acquisition:** Éster Cerdeira Sabino, Lea Campos de Oliveira, Antônio Luiz Pinho Ribeiro.

**Investigation:** Ariela Mota Ferreira, Éster Cerdeira Sabino, Cláudia Di Lorenzo Oliveira, Antônio Luiz Pinho Ribeiro, Maria do Carmo Pereira Nunes, Desirée Sant' Ana Haikal.

**Methodology:** Ariela Mota Ferreira, Éster Cerdeira Sabino, Cláudia Di Lorenzo Oliveira, Clareci Silva Cardoso, Antônio Luiz Pinho Ribeiro, Renata Fiúza Damasceno, Maria do Carmo Pereira Nunes, Desirée Sant' Ana Haikal.

**Project administration:** Ariela Mota Ferreira, Éster Cerdeira Sabino, Lea Campos de Oliveira, Cláudia Di Lorenzo Oliveira, Antônio Luiz Pinho Ribeiro, Desirée Sant' Ana Haikal.

**Resources:** Éster Cerdeira Sabino, Antônio Luiz Pinho Ribeiro, Maria do Carmo Pereira Nunes, Desirée Sant' Ana Haikal.

**Software:** Ariela Mota Ferreira, Éster Cerdeira Sabino, Antônio Luiz Pinho Ribeiro, Maria do Carmo Pereira Nunes, Desirée Sant' Ana Haikal.

**Supervision:** Ariela Mota Ferreira, Éster Cerdeira Sabino, Cláudia Di Lorenzo Oliveira, Clareci Silva Cardoso, Antônio Luiz Pinho Ribeiro, Maria do Carmo Pereira Nunes, Desirée Sant' Ana Haikal.

**Validation:** Ariela Mota Ferreira, Éster Cerdeira Sabino, Antônio Luiz Pinho Ribeiro, Maria do Carmo Pereira Nunes, Desirée Sant' Ana Haikal.

**Visualization:** Ariela Mota Ferreira, Éster Cerdeira Sabino, Lea Campos de Oliveira, Antônio Luiz Pinho Ribeiro, Maria do Carmo Pereira Nunes, Desirée Sant' Ana Haikal.

**Writing – original draft:** Ariela Mota Ferreira, Éster Cerdeira Sabino, Antônio Luiz Pinho Ribeiro, Maria do Carmo Pereira Nunes, Desirée Sant' Ana Haikal.

**Writing – review & editing:** Ariela Mota Ferreira, Éster Cerdeira Sabino, Lea Campos de Oliveira, Cláudia Di Lorenzo Oliveira, Clareci Silva Cardoso, Antônio Luiz Pinho Ribeiro, Renata Fiúza Damasceno, Maria do Carmo Pereira Nunes, Desirée Sant' Ana Haikal.

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
