## [Decision Letter · Decision Letter 0]

3 Feb 2020

Dear Ferreira,

Thank you very much for submitting your manuscript "Impact of the social context on the prognosis of Chagas disease patients: multilevel analysis of a Brazilian cohort" for consideration at PLOS Neglected Tropical Diseases. As with all papers reviewed by the journal, your manuscript was reviewed by members of the editorial board and by several independent reviewers. In light of the reviews (below this email), we would like to invite the resubmission of a significantly-revised version that takes into account the reviewers' comments. 

We cannot make any decision about publication until we have seen the revised manuscript and your response to the reviewers' comments. Your revised manuscript is also likely to be sent to reviewers for further evaluation.

Sincerely,

Walderez O. Dutra, PhD.

Deputy Editor

Alvaro Acosta-Serrano

Deputy Editor

Reviewer's Responses to Questions

**Key Review Criteria Required for Acceptance?**

**Methods**

-Are the objectives of the study clearly articulated with a clear testable hypothesis stated?

-Is the study design appropriate to address the stated objectives?

-Is the population clearly described and appropriate for the hypothesis being tested?

-Is the sample size sufficient to ensure adequate power to address the hypothesis being tested?

-Were correct statistical analysis used to support conclusions?

-Are there concerns about ethical or regulatory requirements being met?

Reviewer #1: Generally it is appropriated, Minor corrections are suggested

Reviewer #2: Introduction section

- Lines 75 and 76. Authors speaks about social problem and social impact, based on data of reference 1. Please, explain what social impact means, because it is not explained into this reference, or alternatively, provide a reference to address it.

- Line 100. Authors said that CD is classically associated with poverty in rural areas (ref 11). I don’t agree with this statement given that in periurban areas of endemic countries, CD transmission (and prevalence) is even higher that in rural areas, due to internal migrant movements. I suggest authors to reformulate the statement

Methods section

- Study design. Why were negatives excluded from the analysis? How do you ensure that also negatives patients did not suffer from the same cardiological affections?

I suggest authors to provide information about this group, if they have. If not, I consider this lack of information a study major limitation. 

- Line 179: Theoretical model and variables. I think that this section should be better organized and included in Study design section

- In this same section (Theoretical model and variables), and even if authors include this information after, I suggest to provide information why (based on which data) AF and QRS complex duration were selected among arrhythmia types and EKG values.

- Line 199: If I understood well, health related behavior variables were asked only at the beginning. From my point of view, it is possible that these value change during the follow-up period. Please provide information of this timepoint evaluation, and if you don’t have it, provide a rational why it was not considered to be included.

- Line 206: why authors selected range 25-75 to dichotomize variables? Based on what? Please, provide more information on this decision.

**Results**

-Does the analysis presented match the analysis plan?

-Are the results clearly and completely presented?

-Are the figures (Tables, Images) of sufficient quality for clarity?

Reviewer #1: Table 2 should be improved

Figures provided in portuguese and in a very poor quality

Reviewer #2: In case of the results, the analysis presented matched with the analysis plan. Results were mainly clearly presented. 

Table 2.

- Regarding self-reporting skin color, and given the high variety and variability of the variable. As a curiosity, why authors decide white/ non-white variables?

- Functional class: At /with limitations/ variable, did authors considered degree limitation? I think that information is scarce as dichotomized in this case. Please provide an explanation why the variable was established this way.

Figure 1 should be translated into English

**Conclusions**

-Are the conclusions supported by the data presented?

-Are the limitations of analysis clearly described?

-Do the authors discuss how these data can be helpful to advance our understanding of the topic under study?

-Is public health relevance addressed?

Reviewer #1: ok

Reviewer #2: - Line 324. 12% of the patients with CD had Cvascular events. Among them, how many were treated? Please, provide information of the analysis of patients by treatment condition

- Lines 376-380. I think that this assumption is not based on the manuscript results presented. To provide information of this assumption was not the objective of the research, so there is no data to support this hypothesis. I suggest authors to reformulate or eliminate it.

- Same comment to 436-438 lines.

- Line 453-454: there were people with altered NTproBNP without cardiovascular events? It seems clear that, as ventricular disfunction marker, proBNP is proven to be higher in older people. Please highlight if new information on this concern regarding this study results

**Editorial and Data Presentation Modifications?**

Reviewer #1: see attached document

Reviewer #2: (No Response)

**Summary and General Comments**

Reviewer #1: see attached document

Reviewer #2: First of all, I would like congratulate researches for the comprehensive view that a disease like Chagas, and that is reflected in the research results and the manuscript.

Nevertheless, and spite of the necessary and positive approach, I would have needed a little more information to arrive to authors’ conclusions.

PLOS authors have the option to publish the peer review history of their article (what does this mean?). If published, this will include your full peer review and any attached files.

Reviewer #1: No

Reviewer #2: No
---

## [Decision Letter · Decision Letter 1]

3 Apr 2020

Dear Ferreira,

Thank you very much for submitting your manuscript "Impact of the social context on the prognosis of Chagas disease patients: multilevel analysis of a Brazilian cohort" for consideration at PLOS Neglected Tropical Diseases. As with all papers reviewed by the journal, your manuscript was reviewed by members of the editorial board and by several independent reviewers. In light of the reviews (below this email), we would like to invite the resubmission of a significantly-revised version that takes into account the reviewers' comments. 

Given the comments by one of the reviewers were not addressed, please consider if you would like to address the comments and resubmit. We will not consider a resubmission without proper response to the points raised.

We cannot make any decision about publication until we have seen the revised manuscript and your response to the reviewers' comments. Your revised manuscript is also likely to be sent to reviewers for further evaluation.

Sincerely,

Walderez O. Dutra, PhD.

Deputy Editor

Alvaro Acosta-Serrano

Deputy Editor

Reviewer's Responses to Questions

**Key Review Criteria Required for Acceptance?**

**Methods**

-Are the objectives of the study clearly articulated with a clear testable hypothesis stated?

-Is the study design appropriate to address the stated objectives?

-Is the population clearly described and appropriate for the hypothesis being tested?

-Is the sample size sufficient to ensure adequate power to address the hypothesis being tested?

-Were correct statistical analysis used to support conclusions?

-Are there concerns about ethical or regulatory requirements being met?

Reviewer #1: Authors have adressed all comments properly

Reviewer #2: Same comments than in R1: there were no answered by authors (nor in the text, neither in the letter of response)

**Results**

-Does the analysis presented match the analysis plan?

-Are the results clearly and completely presented?

-Are the figures (Tables, Images) of sufficient quality for clarity?

Reviewer #1: Authors have adressed all comments properly

Reviewer #2: Same comments than in R1: there were no answered by authors (nor in the text, neither in the letter of response)

**Conclusions**

-Are the conclusions supported by the data presented?

-Are the limitations of analysis clearly described?

-Do the authors discuss how these data can be helpful to advance our understanding of the topic under study?

-Is public health relevance addressed?

Reviewer #1: Authors have adressed all comments properly

Reviewer #2: Same comments than in R1: there were no answered by authors (nor in the text, neither in the letter of response)

**Editorial and Data Presentation Modifications?**

Reviewer #1: Accept

Reviewer #2: No comments

**Summary and General Comments**

Reviewer #1: Obrigado por ter considerado minhas sugestões.

Reviewer #2: No comments

PLOS authors have the option to publish the peer review history of their article (what does this mean?). If published, this will include your full peer review and any attached files.

Reviewer #1: No

Reviewer #2: No
---

## [Decision Letter · Decision Letter 2]

19 May 2020

Dear Ferreira,

We are pleased to inform you that your manuscript 'Impact of the social context on the prognosis of Chagas disease patients: multilevel analysis of a Brazilian cohort' has been provisionally accepted for publication in PLOS Neglected Tropical Diseases.

Best regards,

Walderez O. Dutra, PhD.

Deputy Editor

Alvaro Acosta-Serrano

Deputy Editor

Reviewer's Responses to Questions

**Key Review Criteria Required for Acceptance?**

**Methods**

-Are the objectives of the study clearly articulated with a clear testable hypothesis stated?

-Is the study design appropriate to address the stated objectives?

-Is the population clearly described and appropriate for the hypothesis being tested?

-Is the sample size sufficient to ensure adequate power to address the hypothesis being tested?

-Were correct statistical analysis used to support conclusions?

-Are there concerns about ethical or regulatory requirements being met?

Reviewer #2: Authors answered the most of the queries raised at the first review, and at this moment methods section is pretty clear and complete.

**Results**

-Does the analysis presented match the analysis plan?

-Are the results clearly and completely presented?

-Are the figures (Tables, Images) of sufficient quality for clarity?

Reviewer #2: Authors answered the most of the queries raised at the first review: results are clear and well structured.

**Conclusions**

-Are the conclusions supported by the data presented?

-Are the limitations of analysis clearly described?

-Do the authors discuss how these data can be helpful to advance our understanding of the topic under study?

-Is public health relevance addressed?

Reviewer #2: The conclusions are supported by the data presented, and limitations of the study clearly established.

**Editorial and Data Presentation Modifications?**

Reviewer #2: NA

**Summary and General Comments**

Reviewer #2: I think that the content of the manuscript contains valuable data and information, and add knowledge for better address Chagas Disease with a interdisciplinar view.

PLOS authors have the option to publish the peer review history of their article (what does this mean?). If published, this will include your full peer review and any attached files.

Reviewer #2: No

---

## [Editor Report · Acceptance letter]

22 Jun 2020

Dear Ferreira,

We are delighted to inform you that your manuscript, "Impact of the social context on the prognosis of Chagas disease patients: multilevel analysis of a Brazilian cohort," has been formally accepted for publication in PLOS Neglected Tropical Diseases.

Best regards,

Shaden Kamhawi

co-Editor-in-Chief

Paul Brindley

co-Editor-in-Chief
